# 5G-Based Smart Healthcare and Mobile Network Security: Combating Fake Base Stations

I-Hsien Liu , Meng-Huan Lee, Hsiao-Ching Huang and Jung-Shian Li *

Department of Electrical Engineering, Institute of Computer and Communication Engineering, National Cheng Kung University, Tainan 701401, Taiwan; ihliu@cans.ee.ncku.edu.tw (I.-H.L.); mhlee@cans.ee.ncku.edu.tw (M.-H.L.); hchuang@cans.ee.ncku.edu.tw (H.-C.H.)
* Correspondence: jsli@cans.ee.ncku.edu.tw

**Abstract:** New mobile network technologies, particularly 5G, have spurred a growth in smart healthcare networks. They enable real-time monitoring, personalized treatments, and more. However, these transformative capabilities have also uncovered potential vulnerabilities, emphasizing the urgency to safeguard patient data and healthcare services. This study analyzes the existing research on 5G-based smart healthcare network security with a specific emphasis on fake base station attacks. The research investigates potential security measures to mitigate the impact of fake base station attacks. And based on those findings, we propose a detection scheme to help combat fake base station threats effectively and to avoid the need to install individual apps on smart devices, providing a foothold for future efforts to develop and deploy better countermeasures. To ensure a secure and resilient ecosystem for 5G-based smart healthcare, continuous research and proactive measures are required. By staying vigilant and committed to research and development, we can protect patient privacy, ensure secure data transmission, and enhance the quality of services within smart healthcare networks and other mobile network applications alike.

**Keywords:** 5G; mobile network; smart healthcare; fake base station; information security





## 1. Introduction

The increasing significance of mobile networks has been propelled by the rapid advancements in next-generation communication technologies, notably 5G, and the ongoing digital transformations across diverse sectors. The recent COVID-19 pandemic has further accelerated the demand for telemedicine and remote healthcare solutions, leading to the emergence of innovative paradigms like smart healthcare. Smart healthcare integrates cutting-edge technologies, such as 5G, Internet of Things (IoT) devices, artificial intelligence (AI), and data analytics, to revolutionize healthcare delivery, enabling real-time monitoring, personalized treatments, and improved patient outcomes [1,2].

Figure 1 illustrates an exemplary architecture of a 5G-based smart healthcare network, connecting diverse healthcare services with various devices to deliver seamless healthcare solutions.

However, alongside these transformative capabilities, the widespread adoption of 5G and IoT technologies has also unveiled potential vulnerabilities, especially within smart healthcare networks. A critical concern has arisen in the form of a rise in fake base station attacks, which pose a serious threat to communication security, jeopardizing user privacy and exposing sensitive data to malicious entities. Recent incidents in Taiwan have underscored the urgent need to bolster mobile network security to safeguard users' privacy and to ensure the uninterrupted availability of essential services [3].

This paper investigates the multifaceted security challenges inherent in 5G-based smart healthcare networks, with a specific focus on fake base station attacks. It aims to show how fake base stations can be a threat to smart healthcare networks; to discuss the possible

existing measures; and to develop effective detection measures that uphold resilience and robustness, safeguarding users' privacy and ensuring secure data transmission within a network.

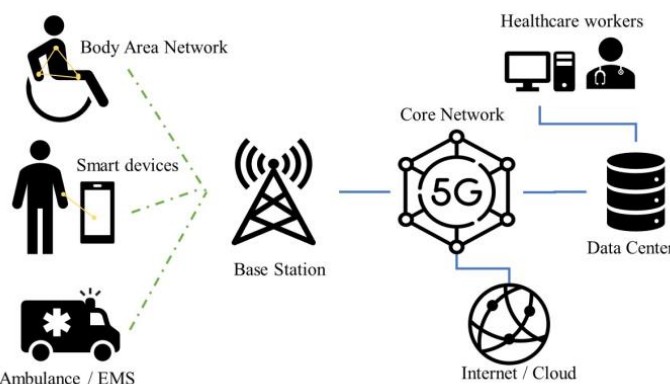

**Figure 1.** An example of a 5G-based smart healthcare network.

## 2. Background

In this chapter, we provide a succinct overview of the 5G mobile network and its integration into smart healthcare networks. Additionally, we explore the existing research on security concerns pertaining to this context.

### 2.1. 5G System and 5G-Based Smart Healthcare Network

The 5G system stands as a cutting-edge mobile network technology designed to offer a low latency, a high bandwidth, and extensive machine-type communication capabilities, catering to a plethora of IoT applications spanning from sensors and robots to medical equipment and wearable electronics. Comprising user equipment (UE), base stations, and a core network, the 5G network forms the foundation of this transformative technology [4].

The UE, serving as a modem, stores a permanent identifier, known as the international mobile subscriber identity (IMSI), and a key essential for mutual authentication between users and the network. Meanwhile, the base station acts as the access point connecting the UE to the radio access network and, subsequently, to the internet. Finally, the core network, functioning as the mobile network's backbone, manages all administrative tasks and traffic routing [4].

Ahad, Tahir, and Yau [5] offer valuable insights into the architecture of 5G- and IoT-enabled smart healthcare networks, emphasizing key enabling technologies such as small cells and software-defined networks. Moreover, they propose a comprehensive taxonomy for 5G smart healthcare, identifying potential future research opportunities within the realm of IoT-based 5G smart healthcare.

Trials of 5G-based smart healthcare networks have been conducted in Taiwanese hospitals, facilitating the deployment of 5G private networks to support medical robots and augmenting medical staff in the wards [6].

### 2.2. Security Concerns of Smart Healthcare

The cyber security of smart healthcare networks stands as a pivotal aspect in ensuring the safe and seamless deployment of critical healthcare services. Protecting sensitive data and personal smart healthcare devices is imperative to safeguard the privacy of patients and medical personnel. Given the mission-critical nature of smart healthcare services, which directly influence human well-being, maintaining high availability and reliability is paramount. Any security breach or service interruption could have life-or-death implications for patients.

Ahad et al.'s research [7] presents an all-encompassing review of 5G-based smart healthcare network security. Their study delves into the technological features and services linked to 5G smart healthcare security, covering aspects such as authentication, confidentiality, availability, nonrepudiation, and integrity. Furthermore, the paper examines various

security threats inherent in 5G smart healthcare connectivity along with the available solutions. It also identifies several research issues and outlines potential future research directions for 5G-based smart healthcare security.

Algarni's research [8] contributes a novel classification scheme for smart healthcare systems and offers insights into the existing literature concerning smart healthcare systems. The study also explores crucial security issues and the countermeasures proposed in the current literature. Lastly, it identifies open research challenges and outlines directions for future research in the domain of smart healthcare systems.

## 3. Fake Base Stations

In this chapter, we delve into the security implications of fake base stations in the context of 5G-based smart healthcare and mobile networks. Base stations play a pivotal role in mobile networks, acting as the termination point for encryption and integrity protection.

Despite their significance, the security of base stations is often overlooked compared to the UE and core network. Nevertheless, securing base stations is critical for safeguarding networks from unauthorized access and potential malicious activities. Base stations act as the first line of defense, enforcing authentication and encryption protocols, monitoring network traffic, and mitigating potential threats. With 5G's advent and increased demand for devices, the density of base stations has substantially risen, necessitating a focus on the recent cases of fake base station attacks.

### 3.1. Fake Base Station Attack

Fake base stations, also known as rouge base stations, are malicious devices that impersonate authentic base stations. Their primary aim is to deceive the UE into connecting to unauthorized services and to potentially extract sensitive user information, such as IMSI data [9]. The presence of these "IMSI-Catchers" has been noted since the 2G era, posing a global concern with reports of suspicious devices appearing in major cities and instances of law enforcement using similar equipment for surveillance [10,11]. And some security researchers have been working on the topic of 5G fake base stations [12].

A recent case in Taiwan involved the use of fake base stations for phishing attacks. The suspects acquired fake base stations from China, allowing them to transmit phishing SMS messages and to trick people into revealing their credit card information. The attackers achieved this by downgrading the victims' phones to 2G, disconnecting them from legitimate 4G/5G base stations, and making them vulnerable to phishing messages [3]. Similar cases have been observed in China, and Yiming Zhang et al. [13] made efforts to characterize these cases, providing insights into the fake base stations' spam ecosystem, which is operated by criminals or unscrupulous businesses.

While recent cases have mainly used fake base stations for phishing attacks targeting credit card information, this demonstrates that fake base stations remain a relevant threat in modern 4G/5G mobile networks. Phishing messages can be easily modified to deceive unsuspecting victims into accessing fake websites, leading to potential privacy breaches and financial losses.

A typical attack procedure using a fake BS attack is illustrated to help understand the attack. In the telecom sector, these activities represent a malicious sequence targeting mobile network vulnerabilities and compromising user security and privacy:

Step 1. Setting up a fake base station:

Malicious actors configure counterfeit hardware and software to mimic genuine base stations, deceiving the targeted cellular network.

Step 2. Entering the target's radio range:

The counterfeit base station is strategically positioned near the victim to lure their mobile device into connecting unwittingly.

Step 3. Performing the desired attacks:

Once the victim's mobile device connects to the spurious base station, attackers can execute a variety of malicious actions designed to compromise the integrity of the targeted network and to harvest sensitive information. These attacks may include the following methods:

- Gathering IMSI and/or location information: Malicious actors exploit the connection to retrieve the IMSI of the victim's device, thereby acquiring a unique identifier associated with the subscriber. Additionally, they may obtain location information, potentially compromising the user's privacy.
- Spam phishing or malicious SMS messages: Attackers leverage the connection to send unsolicited, deceptive, or malicious Short Message Service (SMS) messages to the victim's device. These messages often serve as vehicles for phishing attempts or the delivery of harmful payloads, with the aim of tricking or coercing the user into taking detrimental actions.
- Sending fake reject messages, emergency warnings, etc.: Malicious operators can utilize the fraudulent base station to transmit fabricated network messages, such as fake reject messages or counterfeit emergency alerts. This manipulation can lead to service disruptions or, in the case of false emergency warnings, cause panic and confusion among the affected users.

In summary, this orchestration of a fake base station, strategic placement, and ensuing attacks represents a malevolent intrusion in telecom, posing significant security and privacy risks to mobile network users.

Advanced fake base stations are also capable of performing denial of service (DoS) and man in the middle (MitM) attacks. Moreover, as the cost of deploying mobile networks decreases for legitimate purposes, it also becomes more affordable for malicious actors to obtain the hardware and software required for setting up fake base stations, leading to an increase in low-cost fake base station attacks [14]. Most of the above attacks begin with exploiting vulnerabilities in the attachment and authentication features. Additionally, unprotected unicast messages may leak critical network configurations to attackers, aiding in their malicious endeavors. Fake base station attacks can be coupled with other cyberattacks, such as social engineering, phishing, or exploiting device vulnerabilities, as illustrated by the aforementioned cases.

*3.2. Threats to Smart Healthcare Networks*

Fake base stations pose a significant threat to smart healthcare networks, jeopardizing the confidentiality, integrity, and availability (the CIA triad model) of sensitive healthcare data and services.

- Confidentiality:

A compromised access control and authentication mechanism may allow unauthorized access to healthcare data through fake base stations, leading to the exposure of sensitive patient information. Poorly implemented encryption or its absence may expose system information and user data to eavesdropping, thus infringing on patients' privacy.

- Integrity:

Fake base stations that act as a MitM can intercept and modify information transmitted between devices and legitimate base stations, compromising the integrity of smart healthcare data.

- Availability:

Hacked base stations or fake BSs can cause DoS attacks, severing the connection of health sensors and devices to the authentic network and rendering crucial healthcare services and data unavailable to patients and healthcare providers.

Figure 2 illustrates the security threats posed by fake base stations within smart healthcare networks, underscoring the urgency for effective countermeasures to ensure the

integrity and privacy of patients' data and to maintain the uninterrupted availability of essential healthcare services.

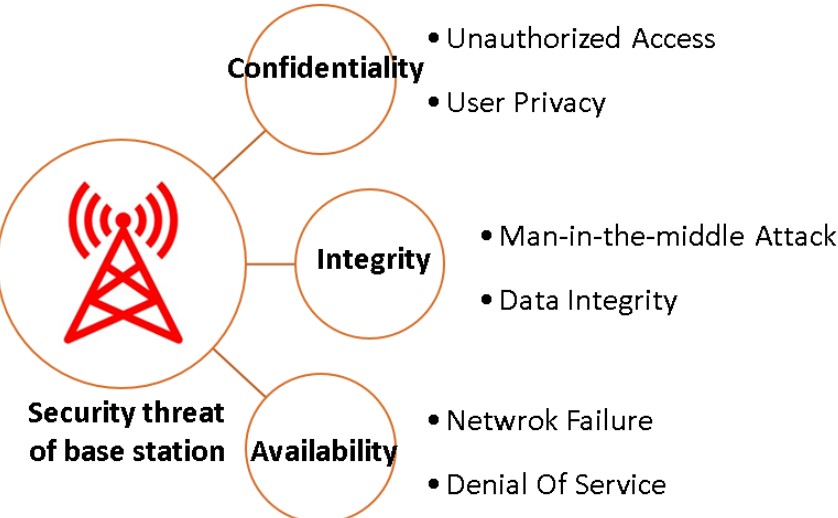

**Figure 2.** Security threat of base station.

## 4. Combating Fake Base Stations

Here, we address the security challenges posed by fake base stations in 5G-based smart healthcare networks and potential security measures to mitigate their impact. We also introduce a detection scheme to identify suspicious base stations and discuss future research directions.

### 4.1. Security Measures

To safeguard smart healthcare networks, one approach is to use private networks dedicated solely to healthcare services. Private networks offer a tailored infrastructure that organizations can control, ensuring higher security levels. However, even private networks can be vulnerable to attacks, especially when sharing resources with the public network infrastructure through network function virtualization (NFV). In such cases, the compromise of shared base stations may impact private network services as well [9].

Leveraging cryptography and blockchain technology presents another promising avenue for network protection. Some research has explored novel approaches using these methods to address security issues. Blockchain technology can serve as a secure and transparent data-sharing platform between patients and healthcare workers, supporting the privacy requirements for diverse 5G applications. Nevertheless, implementing blockchain technology on small IoT devices necessitates substantial computing power, posing certain challenges in resource-constrained environments [15].

Some MNOs also rate-limit attach requests in public networks to protect network resources from getting used up by a huge amount of attach attempts [12]. Also, 3GPP is aware of the security vulnerability of previous generations and started studying security enhancements against fake base stations since Release 15, and the study is continuously documented in a technical report, TR 33.809 [16].

### 4.2. Fake Base Station Detection

Detecting suspicious base stations and potential threats is vital for early identification and proactive action. Take the case in Taiwan; public awareness of the threat of fake base station attacks is still relatively low, and its research and investigation only just started recently because of the recent incident. A big telecom company just released an app with a 2G fake base station detection feature recently. The company is working with government agencies to develop this feature in response to the criminal case mentioned in [3]. Though their detection method is undisclosed, this is still an important step forward against the

fake base station threats in Taiwan [17]. But this kind of traditional detection method requires the installation of special apps on the UE and root privileges to access the modem driver, which are also limited by different hardware constraints and software compatibility issues. UE also comes in various types of devices other than phones, like smart gadgets or health equipment that are difficult to install and maintain apps on individually.

The research community is actively exploring other methods for detecting fake base stations. Network-based, signal-based, and location-based detection schemes have gained considerable attention, offering viable means to counteract fake base station attacks [18,19]. Some of these detection methods can be adapted and applied to the context of 5G and smart healthcare networks.

One of them is the "Crocodile Hunter" project developed by the Electronic Frontier Foundation (EFF) [20]; it is based on open-sourced projects and low-cost hardware, and it aids in detecting fake base stations. We think this is a promising concept for providing detection and protection for 5G-based smart healthcare and mobile networks.

Based on this concept, we designed a detection scheme that resides at the side of the network manager and has the capability to update legitimate cell information from the core network as shown in Figure 3. Our proposed scheme monitors nearby signals, listens to messages broadcasted by base stations in proximity, and detects any suspicious base stations or potentially malicious fake base stations. The detector stores this record in a database to be analyzed by the operators and tracks activities. The detection algorithm continues to monitor the cells and compares them to a whitelist of authentic cell information. The whitelist can be periodically updated to reflect the current configuration of the network and to ensure better security. The detection mechanism is illustrated below as shown in Figure 4.

First, the detector, which is a radio frontend or a device with UE functionalities, performs a cell search and decodes information blocks. Different cell signatures are gathered, including several parameters that fake base station attackers would use, mainly MCC, MNC, PCI, and EARFCN, and the timestamp of when the cells are spotted is also recorded.

After the records are gathered, the detection algorithm examines these cells, and we analyzed the data collected in the database. The system flow of the detection mechanism is shown below. The detection algorithm compares the cell identity with the whitelist and security criteria. The basic criteria were designed based on a whitelist of permitted base stations in the network. The detector sees whether a cell is (a) a real cell that matches the whitelist and belongs to our network, (b) not authorized but belongs to other legitimate operators with legal MCC-MNC, or (c) does not belong to the above criteria and is suspicious.

This proposed detection scheme can serve as a starting point for people who want to build a fake base station detection system without the need to install apps on individual devices.

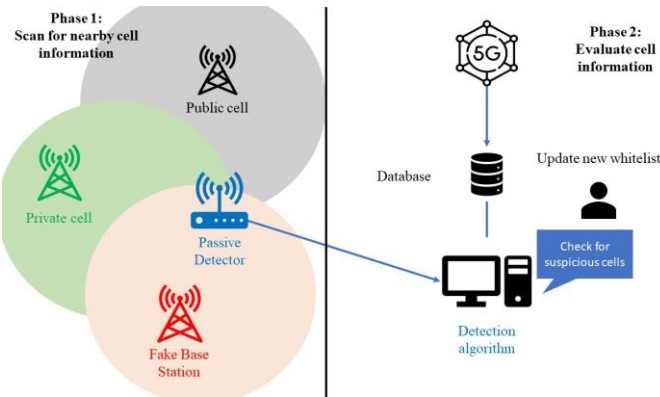

**Figure 3.** Proposed detection scheme.

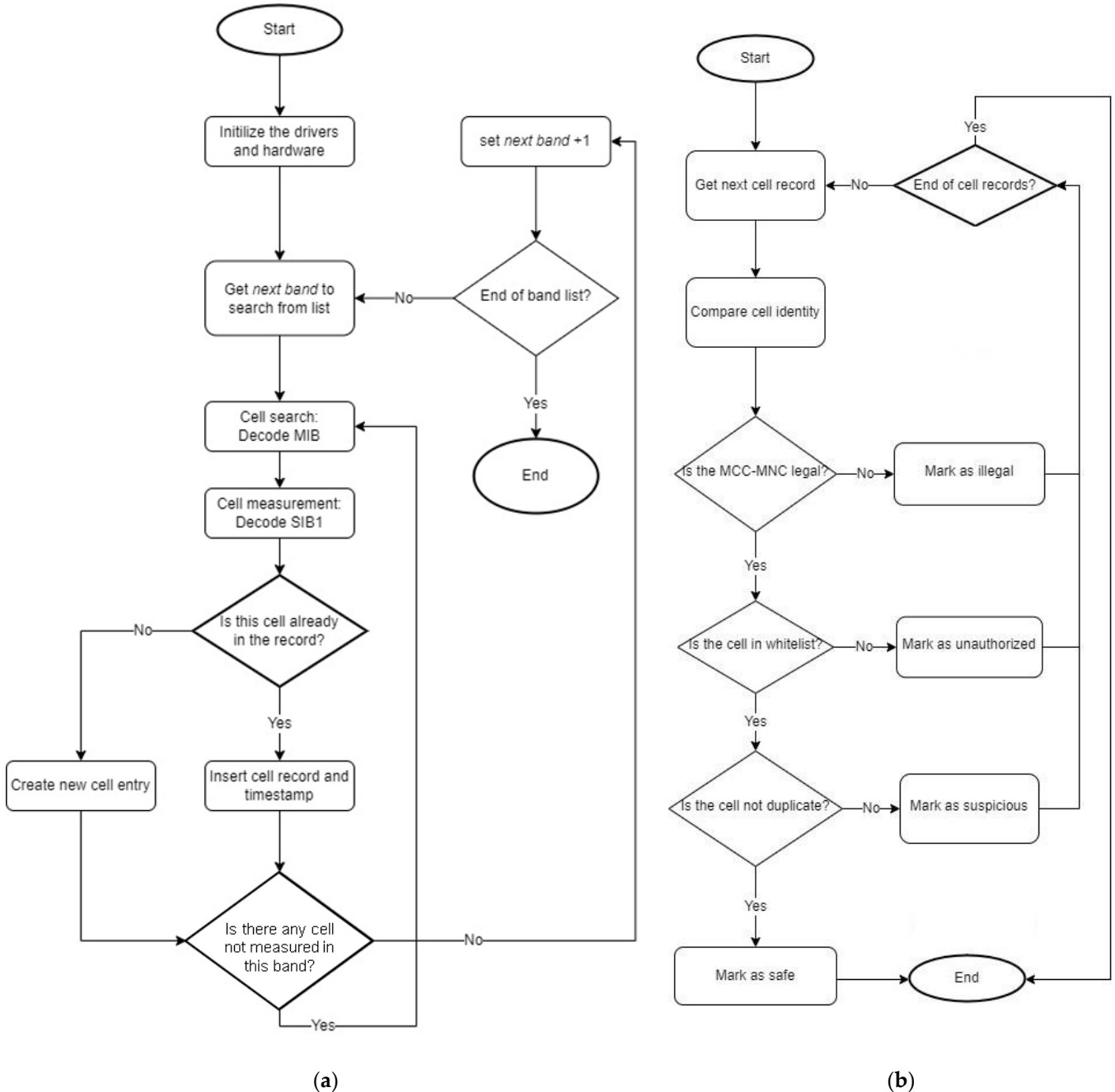

**Figure 4.** Flowcharts of the proposed detection scheme: (**a**) Phase 1—scanning flow chart. (**b**) Phase 2—evaluate cell record.

## 5. Results and Discussion

### 5.1. Detector Deployment

All our experiments were run in our lab at the university. We set up the experiment to gather nearby cell signals to test our concept.

In the first phase, the detector basically did what attackers do during the probing and information-gathering phase, but instead of using this information to perform attacks, we stored this information to analyze the different cell identities for further use. The detector scanned through the bands of a band list that contained all the bands that needed to be scanned. A simple loop script could be used to repeatedly run the *cell_search* and *cell_mesaurement* program, iterating through each band EARFCN number as the parameters. We gathered the different configurations and parameters of the fake base stations and real

base stations from the previous attack phase. The public network data are real signals broadcasted by MNOs that were received in our lab using *Crocodilehunter*. We edited the config.ini to reflect the network of Taiwan, setting the expected MCC to 01 and 466. Table 1 shows some of the entries of the records. An MCC (mobile country code) is a three-digit numeric code that is allocated by the ITU and uniquely identifies a specific country. An MNC (mobile network code) is used to identify a specific mobile network operator within the country. A TAC (tracking area code) enables finer-grained tracking and management within a network's coverage areas. These codes work together to ensure efficient and accurate mobile network operations and subscriber management.

**Table 1.** Detector cell records.

| eNB ID | PCI | MCC | MNC | EARFCN | TAC | Last Seen |
|--------|-----|-----|-----|--------|-----|-----------|
| 701115 | 460 | 466 | 97 | 275 | 44280 | 17 January 2023 15:45 |
| 176827 | 151 | 466 | 97 | 3050 | 60664 | 11 January 2023 11:08 |
| 186013 | 157 | 466 | 01 | 50 | 3408 | 16 January 2023 02:47 |
| 579485 | 157 | 466 | 03 | 50 | 23888 | 9 January 2023 18:07 |

In the second phase, the cell record went through an analysis process to be examined. We constructed a whitelist of "real cells" as a reference, that is, using the appropriate MCC-MNC. We analyzed the records gathered in the previous steps. During a period of 7 days, we discovered that the scanner scanned 33,579 times and picked up a total of 145 distinct cell entries. Most of them were public network cells starting with MCC 466 as shown in Table 2. However, we discovered multiple cells—121, to be exact—that did not belong to MCC 466 or 01 as shown in Table 3. But they were indeed present; we tried to look up those unknown MCCs, but there are no corresponding country code–network code combinations in Taiwan as far as we know.

**Table 2.** Legal MCC-MNC entries that were filtered out.

| MCC | MNC | TAC | PhyID | EARFCN | Time Stamp | Frequency | eNodeB_ID |
|-----|-----|-----|-------|--------|-----------|-----------|-----------|
| 466 | 97 | 44280 | 151 | 275 | 17 January 2023 15:35 | 2137.5 | 701115 |
| 466 | 1 | 19792 | 292 | 3250 | 17 January 2023 15:35 | 2670 | 710023 |
| 466 | 1 | 19792 | 292 | 3250 | 17 January 2023 15:34 | 2670 | 710023 |
| 466 | 1 | 19792 | 151 | 275 | 17 January 2023 15:34 | 2137.5 | 710023 |
| 466 | 1 | 19792 | 61 | 3050 | 17 January 2023 15:33 | 2650 | 710023 |
| 466 | 1 | 19792 | 151 | 275 | 17 January 2023 15:32 | 2137.5 | 710023 |
| 466 | 1 | 19792 | 47 | 9385 | 17 January 2023 15:31 | 775.5 | 710023 |
| 466 | 1 | 19792 | 292 | 3250 | 17 January 2023 15:31 | 2670 | 710023 |
| 466 | 1 | 19792 | 47 | 9385 | 17 January 2023 15:31 | 775.5 | 710023 |
| 466 | 97 | 44280 | 292 | 3250 | 17 January 2023 15:31 | 2670 | 701115 |
| 466 | 97 | 44280 | 151 | 275 | 17 January 2023 15:30 | 2137.5 | 701115 |

**Table 3.** Some unknown MCC-MNC entries that were filtered out.

| MCC | MNC | TAC | PhyID | EARFCN | Time Stamp | Frequency | eNodeB_ID |
|-----|-----|-----|-------|--------|-----------|-----------|-----------|
| 566 | 17 | 44140 | 151 | 275 | 11 January 2023 12:03 | 2137.5 | 692923 |
| 566 | 17 | 44140 | 151 | 275 | 11 January 2023 12:02 | 2137.5 | 692923 |
| 476 | 97 | 44280 | 151 | 275 | 11 January 2023 07:50 | 2137.5 | 701115 |

**Table 3.** *Cont.*

| MCC | MNC | TAC | PhyID | EARFCN | Time Stamp | Frequency | eNodeB_ID |
|---|---|---|---|---|---|---|---|
| 476 | 97 | 44280 | 151 | 275 | 11 January 2023 07:49 | 2137.5 | 701115 |
| 447 | 41 | 19792 | 157 | 50 | 11 January 2023 06:03 | 2115 | 694173 |
| 447 | 41 | 19792 | 157 | 50 | 11 January 2023 06:01 | 2115 | 694173 |
| 666 | 0 | 19792 | 157 | 50 | 11 January 2023 01:43 | 2115 | 710557 |
| 666 | 0 | 19792 | 157 | 50 | 11 January 2023 01:42 | 2115 | 710557 |
| 465 | 95 | 48376 | 292 | 3250 | 11 January 2023 01:17 | 2670 | 702139 |
| 566 | 17 | 44140 | 460 | 275 | 11 January 2023 12:03 | 2137.5 | 692923 |
| 566 | 17 | 44140 | 151 | 275 | 11 January 2023 12:03 | 2137.5 | 692923 |

*5.2. Discussion*

Although our detection scheme successfully filtered these cells' signals out because they did not match our whitelist or the legal MCC-MNC configurations, this does not mean that they are all malicious base stations that are intended to perform attacks. After discussing with the related authorities about this result, we suspect that there could be several reasons for it: First, some network operators will share the same cell to reduce the physical cost of maintaining the hardware equipment. Second, during the long period of scanning, there might be errors that occur when decoding the packets since our setup is lacking an external GPS clock to provide a more precise timing signal and has to rely on the internal clock on the SDR. Thirdly is that there might just be a testing/experimenting signal coming from another laboratory nearby the campus. Therefore, we cannot conclude whether or not these unknown base stations are all malicious, but we did treat them as illegal.

The results showed that the detection scheme is feasible, without the need to install dedicated apps on the UE devices. And the data collected can be used to analyze and filter out nearby cells according to the whitelist configuration. We also discovered the caveat of our current design: for its large-scale deployment in a real-world mobile network, one may need to consider keeping track of the roaming networks or other guest networks and identifying and adding them to the detection criteria. And multiple operators can share important information, like helping maintain the whitelist together and forming an alliance, to maximize the effect of this detection scheme.

When an unknown or suspicious base station is reported by the detector, we advise that the operators can further respond in several ways. First, they should further investigate the signal by other means, either a different detector or other detection methods, and check the detailed record, comparing it to the up-to-date information of their own network as well as notifying other operators and/or the authorities. Also, one should watch for any patterns over a period of time, preferably across multiple records, to make better decisions.

**6. Conclusions**

In summary, the advent of 5G-based smart healthcare networks presents unparalleled opportunities for healthcare's transformation. However, these advancements also bring forth significant security challenges that require immediate attention. Notably, fake base station attacks pose a critical threat, compromising communication security, user privacy, and sensitive healthcare data. This paper delved into the multifaceted security concerns within 5G-based smart healthcare networks, with a specific focus on fake base station attacks. It underscored the importance of implementing robust security measures, such as private networks and blockchain-based solutions. Additionally, we introduced a detection mechanism for identifying suspicious base stations and discussed potential avenues for future research.

Regarding the mechanism for identifying suspicious base stations, it relies on MCC-MNC values. While our detection system efficiently filters out signals from these rogue cells due to their nonconformity with the whitelist and legitimate MCC-MNC configurations, determined attackers may seek to collect pertinent information beforehand to manipulate their transmission parameters. Given the inherently illicit nature of these attacks, practical considerations suggest that, when the detector reports unknown or suspicious base stations, the network operators should respond through various means.

In the future, the detection scheme proposed in this research could be integrated with other methods, such as GPS-based signal source location as proposed by other researchers, to assist in identifying rogue base stations or in utilizing PCI information for automatic neighbor relations [21], among other approaches. We encourage readers to propose alternative methods or platforms for implementing and enhancing the proposed solution. Considering diverse real-world applications, including IoT use cases, beyond just smart healthcare is essential.

Furthermore, it is advisable to conduct further signal analysis using diverse detectors or alternative detection methods. The comprehensive scrutiny of detailed records in comparison with the most up-to-date network information is recommended. If possible, the continuous observation of patterns over time, ideally across multiple records, provides a more comprehensive understanding of the situation.

**Author Contributions:** Validation, H.-C.H.; Writing—original draft, M.-H.L.; Writing—review & editing, I.-H.L.; Supervision, J.-S.L. All authors have read and agreed to the published version of the manuscript.

**Funding:** This work was supported by the National Science and Technology Council (NSTC) in Taiwan under contract numbers 111-2221-E-006-079- and 112-2634-F-006-001-MBK.

**Institutional Review Board Statement:** Not applicable.

**Informed Consent Statement:** Not applicable.

**Data Availability Statement:** Data available on request due to restrictions eg privacy or ethical. The data presented in this study are available on request from the corresponding author. The data are not publicly available due to the dataset containing base station signal records that do not comply with regulatory approvals.

**Conflicts of Interest:** The authors declare no conflict of interest.

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
