# Peer review of "5G-Based Smart Healthcare and Mobile Network Security: Combating Fake Base Stations"

_applsci, doi:10.3390/app132011565_

Round 1

Reviewer 1 Report

I have read this article multiple times and have been unable to determine whether it is a communication, research, or review article. In addition, author affiliations are not disclosed other than email addresses, suggesting that the authors were in a haste to submit the article.

I recommend that the authors resubmit this article after reading the author's guidelines for this journal so that I may be able to review it technically.

Best wishes

OK

Author Response

We appreciate your comments, and we apologize for the oversight in not specifying the affiliations of the authors during the revision process. All the authors of this paper are affiliated with the Department of Electrical Engineering and the Institute of Computer and Communication Engineering at National Cheng Kung University, Taiwan.

This article falls under the Communication type paper and is aimed at addressing the actual challenges posed by rogue base stations in the current 5G telecommunication environment. It presents a viable emergency solution in response to this issue.

Author Response

Thank you for your suggestions. Regarding the relevant recommendations you provided, the corresponding revisions have been made. The summary of the revisions is as follows:

1. Regarding the possibility of fake base stations emitting legitimate MCC-MNC values, this potential scenario has been addressed in the second paragraph of the "6. Conclusion" section. A relevant discussion and recommendation have been included, as follows:
In this paper, the mechanism for identifying suspicious base stations is based on the MCC-MNC values. Although our detection system effectively filters out signals from these cells due to their mismatch with the whitelist and legitimate MCC-MNC configurations, determined attackers may attempt to pre-collect relevant information to adjust their transmission parameters. Given the inherently illicit nature of such attacks, practical con-siderations suggest that when the detector reports unknown or suspicious base stations, network operators are advised to respond through multiple means. Furthermore, it is ad-visable to conduct further signal analysis using diverse detectors or alternative detection methods. Detailed records should be reviewed and compared with the most up-to-date network information. If possible, continuous observation of any patterns over a period, ideally across multiple records, is recommended to obtain more comprehensive insights.

2. While the source of inspiration, "Crocodile Hunter," has ceased updates, it remains accessible for scholars interested in subsequent research. Furthermore, alternative methods such as Open UE or SDR can be used for related research experiments.

3. For clarification on abbreviations such as MCC, MNC, and TAC, relevant explanations have been added in the paragraph preceding Table 1. A portion of the added text reads as follows:
MCC (Mobile Country Code) is a three-digit numeric code that is allocated by the ITU and uniquely identifies a specific country. MNC (Mobile Network Code) is used to identify a specific mobile network operator within the country. TAC (Tracking Area Code) enables finer-grained tracking and management within the network's coverage areas. These codes work together to ensure efficient and accurate mobile network operations and subscriber management.

4. In this paper, considering the demand for Smart Healthcare and the development of 5G private networks, the potential risks associated with the presence of fake base stations have been discussed. Products like RESMED's positive airway pressure devices have begun integrating and utilizing telecommunications network technologies for healthcare data collection. However, as specific use cases are limited and not explicitly focused on attacks against these products, specific Smart Healthcare examples have not been explicitly named in the paper.

Reviewer 3 Report

Paper objective is good

References to be improvised . Also shall cite papers. Include recent references.

Comparison table is missing.

updated with new flow of work instead of flow chart

check grammer throughout the paper

Paper objective is good

References to be improvised . Also shall cite papers. Include recent references.

Comparison table is missing.

updated with new flow of work instead of flow chart

check grammer throughout the paper

Author Response

We appreciate your suggestions, and we have made the necessary revisions in response to your feedback. However, due to the limited availability of relevant literature on fake base stations, we have only incorporated a recent reference, Ref 21, published in 2022. This reference has been added, and a brief discussion has been included in the conclusion section as follows:
In the future, the detection scheme proposed in this research can be integrated with other methods, such as GPS-based signal source location proposed by other researchers, to assist in identifying rogue base stations or utilizing PCI information for Automatic Neighbour Relation [21], among other approaches. We encourage readers to propose al-ternative methods or platforms for implementing and enhancing the proposed solution. Considering diverse real-world applications, including IoT use cases, beyond just smart healthcare, is essential.

Reviewer 4 Report

All abbreviations must be verified. For example, "(IMSI)" and "(FBS)" are abbreviated twice, but "(SDN)" and "(RAN)" are not used in the text.
Figure 2 contains test information only. It is better to replace the drawing with dough.
In Figure 3, element names are in smaller font than comments. The figure does not contain new information. You can do without it.
There is one final state missing in Figure 5b.
Section 4 ends with a picture, but there should be text.
Information about future research is best placed in the conclusions.

There is no experiment in the article. This paper is of an engineering rather than scientific nature. The algorithm cannot be a scientific result, and no metrics or criteria for assessing security are provided.

Author Response

We appreciate your comments. Regarding the suggestions you provided, we have made the following revisions:

The presentation of abbreviations has been adjusted once again, and unnecessary abbreviations have been removed.

Figure 2 has been replaced with itemized content, eliminating the need for the figure.

Figure 3 (now Figure 2) has been reorganized for better font clarity according to hierarchy.

The final state in Figure 5b (now Figure 4b) has been included.

The discussion of future research has been relocated to the conclusion section.

Round 2

Reviewer 1 Report

The revised version of the article has resulted in notable improvements. However, the conclusions section requires revisions. Also, future work should also be highlighted.

Acceptable

Author Response

Thank you for your suggestions. The conclusions section has been revised accordingly, and the modifications have been highlighted in red within the manuscript.

Reviewer 4 Report

The abbreviation is indicated once at the first mention, and then only the abbreviation is used (except for names in headings and images). In this case, the problem with abbreviations remains, for example, “User Equipment (UE)”, “Mobile Subscriber Identity (IMSI)”, “PCI”, etc. It is advisable to correct everything before publication.

Author Response

Thank you for your suggestions. Regarding the issue of word abbreviations, we have made the necessary revisions, and the modified sections have been indicated in red in the manuscript.